# GraphCL: Graph-based Clustering for Semi-Supervised Medical Image Segmentation

**Mengzhu Wang** [1]  **Houcheng Su** [2]  **Jiao Li** [3]  **Chuan Li** [4]  **Nan Yin** [2]  **Li Shen** [5]  **Jingcai Guo** [6]

## Abstract

Semi-supervised learning (SSL) has made notable advancements in medical image segmentation (MIS), particularly in scenarios with limited labeled data and significantly enhancing data utilization efficiency. Previous methods primarily focus on complex training strategies to utilize unlabeled data but neglect the importance of graph structural information. Different from existing methods, we propose a graph-based clustering for semi-supervised medical image segmentation (GraphCL) by jointly modeling graph data structure in a unified deep model. The proposed GraphCL model enjoys several advantages. Firstly, to the best of our knowledge, this is the first work to model the data structure information for semi-supervised medical image segmentation (SSMIS). Secondly, to get the clustered features across different graphs, we integrate both pairwise affinities between local image features and raw features as inputs. Extensive experimental results on three standard benchmarks show that the proposed GraphCL algorithm outperforms state-of-the-art semi-supervised medical image segmentation methods. The source code is available at https://github.com/dreamkily/GraphCL

## 1. Introduction

Medical image segmentation derives from computed tomography (CT) (Buzug, 2011) or magnetic resonance imaging (MRI) (Glover, 2011), plays a crucial role in various clinical applications (Tang et al., 2021a;b; Wang et al., 2024b;c;a).

[1]Hebei University of Technology [2]Hong Kong University of Science and Technology [3]University of Electronic Science and Technology of China [4]National University of Defense Technology [5]Sun Yat-Sen University [6]The Hong Kong Polytechnic University. Correspondence to: Nan Yin <yinnan8911@gmail.com>, Jingcai Guo <jc-jingcai.guo@polyu.edu.hk>.

*Proceedings of the 42nd International Conference on Machine Learning*, Vancouver, Canada. PMLR 267, 2025. Copyright 2025 by the author(s).

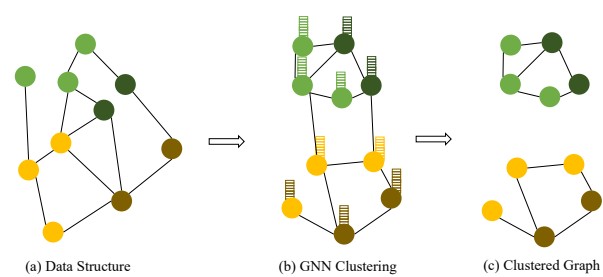

(a) Data Structure    (b) GNN Clustering    (c) Clustered Graph

*Figure 1.* We apply graph neural networks (GNN) to address SSMIS challenges. Specifically, we create a graph representation to capture the data structure, and then use GNN-based clustering to group the graphs.

However, obtaining large medical datasets with precise labels for training segmentation models is challenging, as a substantial amount of labeled images can only be provided by experts. This significantly limits the development of medical image segmentation algorithms and poses substantial challenges for further research due to the scarcity of labeled data. To address these challenges, semi-supervised medical image segmentation (SSMIS) (Bai et al., 2023; Wu et al., 2022; Shi et al., 2021) has emerged as an effective approach, enabling segmentation models to learn from a small set of labeled examples in conjunction with easily accessible unlabeled data.

In the field of SSMIS, although labeled and unlabeled data are theoretically expected to come from the same distribution, in practice, due to the extremely limited availability of labeled data, it is challenging to accurately infer the true distribution of the data. This often leads to the issue of distribution mismatch between the large amount of unlabeled data and the small set of labeled data (Wang et al., 2019). To address this challenge, several SSMIS techniques have been developed. For example, BCP (Bai et al., 2023) enhances data consistency by randomly cropping regions in labeled images (as foreground) and pasting them onto unlabeled images (as background), and vice versa. TI-ST (Ghamsarian et al., 2023) proposes a semi-supervised learning strategy called transformation-invariant self-training, which

improves domain adaptation by evaluating the reliability of pixel-level pseudo-labels and excluding unreliable predictions during the self-training process. ContrastMask (Wang et al., 2022) implements dense contrastive learning on both labeled and unlabeled data. Recent advancements in the field have shown promising results, especially in traditional graph theory applications such as object detection (Zhao et al., 2021; Song et al., 2023) and tracking (Li et al., 2020a; Hyun et al., 2023). However, all of these methods overlook the role of graph information in enhancing semi-supervised medical image segmentation.

Recently, GraphNet (Pu et al., 2018) make an innovative attempt in the supervised semantic segmentation field by applying Graph Convolutional Networks (GCN)(Kipf & Welling, 2016; Wang et al., 2025) to the task. The researchers transformed images into unweighted graph structures by aggregating pixels from superpixel techniques into graph nodes(Achanta et al., 2012). These graphs were then fed into a standard GCN equipped with a cross-entropy loss function to generate pseudo-labels. $A^2$GNN (Zhang et al., 2021) propose an innovative affinity-based convolutional neural network capable of converting images into weighted graph forms. While these graph neural network-based methods have shown outstanding performance in traditional image tasks, they have received limited attention in the medical image domain. Medical image segmentation presents unique challenges, such as complex biological structures and high sensitivity to pathological changes. Moreover, no research has explored SSMIS specifically from the perspective of data structure.

To address this issue, we propose a graph-based clustering for semi-supervised medical image segmentation (GraphCL) by jointly modeling graph data structure in a unified deep model. When modeling data structures in the deep learning network, we create a dense instance graph reflecting the structural similarity of the samples based on CNN features. Each node in the graph corresponds to the CNN features of a sample, which are extracted using a standard convolutional neural network. Then, we deploy a Graph Convolutional Network (GCN) on this instance graph, allowing the structural information to be propagated through learnable weighted edges in the network design. To further improve segmentation accuracy, we introduce a $k$-less clustering strategy that eliminates the need to specify the number of clusters $k$, enabling similar nodes to automatically form clusters (see Figure 1). This strategy significantly enhances the flexibility and adaptability of the model. The core contributions of this study are summarized as follows:

- We propose a graph-based clustering for semi-supervised medical image segmentation by modeling data structure in a unified network. To the best of our knowledge, this is the first work to model the data

structure information in graph for SSMIS.

- We design a graph clustering objective as a loss function to optimize the correlation clustering task in SSMIS.

- Extensive experiments on popular medical image segmentation benchmarks show that GraphCL achieves superior performance.

## 2. Method

### 2.1. Notations and Definitions

In medical image segmentation (MIS), a 3D volume is represented as $\mathbf{X} \in \mathbb{M}^{C \times W \times H \times D}$, where $C$, $W$, $H$, and $D$ correspond to the channel, width, height, and depth, respectively. The goal of semi-supervised segmentation is to predict a pixel-wise label map $\hat{\mathbf{Y}} \in \{0, 1, \ldots, k-1\}^{C \times W \times H \times D}$, indicating the distribution of background and target classes, with $k$ representing the number of classes. The training set $S$ comprises labeled data $A$ and a much larger unlabeled dataset $B$, such that $S = S_l \cup S_u$, where $S_l = \{(\mathbf{X}_i^l, \mathbf{Y}_i^l)\}_{i=1}^A$ is the labeled subset, and $S_u = \{\mathbf{X}_j^u\}_{j=A+1}^{A+B}$ is the unlabeled subset.

During training, we generate mixed samples by selecting two labeled images $(\mathbf{X}_j^l, \mathbf{X}_k^l)$ and two unlabeled images $(\mathbf{X}_m^u, \mathbf{X}_n^u)$. A foreground region is randomly cropped from $\mathbf{X}_j^l$ and pasted onto $\mathbf{X}_n^u$ to produce the mixed image $\mathbf{X}^{\text{out}}$, while another crop from $\mathbf{X}_m^u$ is pasted onto $\mathbf{X}_k^l$ to form $\mathbf{X}^{\text{in}}$. These mixed samples allow the network to learn comprehensive semantic information, leveraging both inward ($\mathbf{X}^{\text{in}}$) and outward ($\mathbf{X}^{\text{out}}$) perspectives.

Our method is built upon a teacher-student framework (Bai et al., 2023), where both networks adopt an encoder-decoder architecture. In the encoder, we incorporate a structure-aware graph network to explicitly capture structural relationships between the inward and outward images. This graph network models spatial and semantic correlations, improving the model's ability to understand the inherent structures within medical images. Additionally, the extracted features dynamically optimize the graph structure, refining cluster assignments to better represent shared and distinct characteristics across labeled and unlabeled data. Finally, both $\mathbf{X}^{\text{in}}$ and $\mathbf{X}^{\text{out}}$ are passed through the student network to predict the segmentation masks $\hat{\mathbf{Y}}^{\text{in}}$ and $\hat{\mathbf{Y}}^{\text{out}}$, which are supervised by the teacher network's predictions on the unlabeled images and the ground truth labels from the labeled images. The pipeline of our proposed method is illustrated in Figure 2.

### 2.2. Bidirectional Copy-Paste Framework

The Bidirectional Copy-Paste (BCP) framework integrates a teacher network $\mathcal{T}(\mathbf{X}_m^u, \mathbf{X}_n^u; \Theta_t)$ and a student network

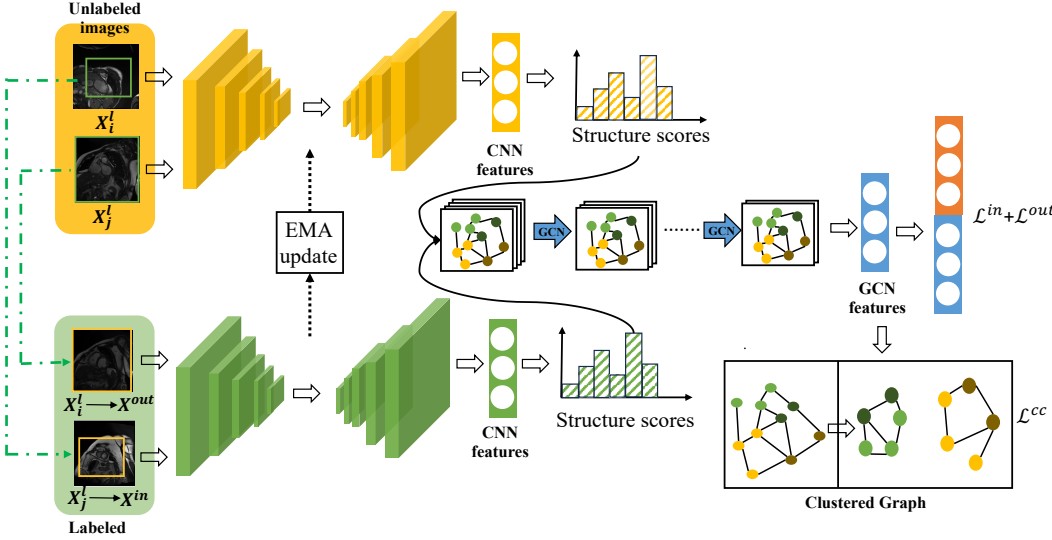

**Figure 2.** The proposed Graph-based Clustering for Semi-supervised Medical Image Segmentation (GraphCL) architecture consists of two core graph mechanisms: GCN alignment and clustering graph construction. In the GCN alignment phase, a data structure analysis network generates structured scores containing structural information, while the CNN is responsible for feature extraction. These structured scores are combined with the CNN-extracted features to construct a dense instance graph for the GCN. After merging the features from both CNN and GCN, the system inputs them to align data from the same category. Regarding the clustering graph construction, we create a similarity matrix based on the similarity between local features, which is then used as the adjacency matrix for the graph. Finally, we utilize this adjacency matrix and deep features as node features to complete the graph construction.

$\mathcal{S}(\mathbf{X}^{\text{in}}, \mathbf{X}^{\text{out}}; \Theta_s)$ (Tarvainen & Valpola, 2017) to enhance semi-supervised medical image segmentation (SSMIS) through a coordinated training strategy, where $\Theta_t$ and $\Theta_s$ are parameters. Initially, the student network is pre-trained using only labeled data to build a supervised model and teacher network leverage the pre-trained model to generate pseudo-labels for unlabeled data during self-training. In the pre-training phase, we adapt the following strategy:

$$\hat{Y}_m^u = \mathcal{T}(X_m^u, \Theta_t) \tag{1}$$

$$\hat{Y}_n^u = \mathcal{T}(X_n^u, \Theta_t) \tag{2}$$

where $X_m^u$ and $X_n^u$ are the unlabeled images, and $\hat{Y}_m^u, \hat{Y}_n^u$ are the corresponding probability maps. The pseudo-labels are initialized using thresholding or argmax operations, depending on whether the task involves binary or multi-class segmentation.

In the bidirectional supervision phase, mixed images $X^{\text{in}}$ and $X^{\text{out}}$ are constructed using a mask $\mathcal{M} \in \{0, 1\}^{C \times W \times H \times D}$, indicating whether a voxel originates from the foreground or background. These images are generated as follows:

$$X^{\text{in}} = X_j^l \odot \mathcal{M} + X_m^u \odot (1 - \mathcal{M}) \tag{3}$$

$$X^{\text{out}} = X_n^u \odot \mathcal{M} + X_k^l \odot (1 - \mathcal{M}) \tag{4}$$

where $X_j^l$ and $X_k^l$ are labeled images, and $\odot$ denotes element-wise multiplication. The corresponding pseudo-labels and ground truth labels are then combined to supervise the student network using the following supervisory signals:

$$Y^{\text{in}} = Y_j^l \odot \mathcal{M} + \hat{Y}_m^u \odot (1 - \mathcal{M}) \tag{5}$$

$$Y^{\text{out}} = \hat{Y}_n^u \odot \mathcal{M} + Y_k^l \odot (1 - \mathcal{M}) \tag{6}$$

We use a weight $\alpha$ to control the contribution of unlabeled image pixels to the loss function. The loss functions for $\mathbf{X}^{\text{in}}$ and $\mathbf{X}^{\text{out}}$ are computed respectively by

$$\mathcal{L}^{\text{in}} = \mathcal{L}_{\text{seg}}\left(\mathbf{Q}^{\text{in}}, \mathbf{Y}^{\text{in}}\right) \odot \mathcal{M} + \alpha \mathcal{L}_{\text{seg}}\left(\mathbf{Q}^{\text{in}}, \mathbf{Y}^{\text{in}}\right) \odot (1 - \mathcal{M}) \tag{7}$$

$$\mathcal{L}^{\text{out}} = \mathcal{L}_{\text{seg}}\left(\mathbf{Q}^{\text{out}}, \mathbf{Y}^{\text{out}}\right) \odot (1 - \mathcal{M}) + \alpha \mathcal{L}_{\text{seg}}\left(\mathbf{Q}^{\text{out}}, \mathbf{Y}^{\text{out}}\right) \odot \mathcal{M} \tag{8}$$

where $\mathcal{L}_{\text{seg}}$ is the linear combination of dice loss and cross-entropy loss. $\mathbf{Q}^{\text{in}}$ and $\mathbf{Q}^{\text{out}}$ are computed by:

$$\mathbf{Q}^{\text{in}} = \mathcal{S}_s\left(\mathbf{X}^{\text{in}}; \Theta_s\right), \quad \mathbf{Q}^{\text{out}} = \mathcal{S}_s\left(\mathbf{X}^{\text{out}}; \Theta_s\right) \tag{9}$$

At each iteration, we update the parameters $\Theta_s$ in the student network by stochastic gradient descent with the loss function:

$$\mathcal{L}_{\text{all}} = \mathcal{L}^{\text{in}} + \mathcal{L}^{\text{out}} \tag{10}$$

Afterwards, teacher network parameters $\Theta_t^{(k+1)}$ at the $(k+1)$-th iteration are updated:

$$\Theta_t^{(k+1)} = \lambda\Theta_t^{(k)} + (1-\lambda)\Theta_s^{(k)} \tag{11}$$

where $\lambda$ is the smoothing coefficient parameter.

## 2.3. Structural Graph Model for Segmentation

In fact, existing studies (Zhou et al., 2023; Robert et al., 2023; Huang et al., 2023) focus on modeling the data structure information for semantic segmentation and have achieved remarkable success, which further emphasizes the critical role of data structure information. To effectively model data structures in semi-supervised medical image segmentation (SSMIS), we propose a graph-based clustering for semi-supervised medical image segmentation (GraphCL).

To address the challenge of effectively integrating both labeled and unlabeled medical images within the semi-supervised medical image segmentation (SSMIS) framework, we propose a Structural Graph Model (SGM). The model is based on the idea that the spatial and semantic structure of images can be efficiently abstracted into a graph, where nodes represent the features of image regions, and edges describe the relationships between these regions. This graph structure enables flexible information propagation, which is crucial for enhancing segmentation accuracy, particularly when labeled data is limited.

### 2.3.1. Structure-aware Alignment

In our graph construction framework, each sample in a mini-batch is treated as a node, and the relationships between nodes are modeled using a Data Structure Analyzer (DSA). This component generates structure scores that quantify the similarity between different samples based on their internal spatial structure, as derived from the learned CNN features. Formally, the feature extraction process for each 3D medical image $\mathbf{X}_{\text{batch}}$ is expressed as:

$$\mathbf{X} = \text{CNN}(\mathbf{X}_{\text{batch}}) \tag{12}$$

where $\mathbf{X}$ represents the graph signal, encoding the features of individual samples. These features are crucial in defining the graph adjacency matrix $\hat{\mathbf{A}}$, computed as:

$$\hat{\mathbf{A}} = \mathbf{X}_{\text{sa}}\mathbf{X}_{\text{sa}}^{\top} \tag{13}$$

where $\mathbf{X}_{\text{sa}} \in \mathbb{R}^{w \times h}$ are the structure scores output by the DSA network, with $w$ denoting the batch size and $h$ the dimension of the structure features. The intuition is that samples with similar structural characteristics should have stronger connections in the graph, which facilitates effective feature propagation during the segmentation task.

Once the graph is constructed, we employ a Graph Convolutional Network (GCN) to perform feature propagation across

nodes. The GCN operates on the instance graph, with the goal of refining the feature representations by aggregating information from neighboring nodes, thereby capturing both local and global structural patterns within the mini-batch. The graph convolution is performed as:

$$\mathbf{Z} = \hat{\mathbf{D}}^{-\frac{1}{2}}\hat{\mathbf{A}}\hat{\mathbf{D}}^{-\frac{1}{2}}\mathbf{X}\mathbf{W} \tag{14}$$

where $\mathbf{Z}$ is the output feature matrix, $\mathbf{W}$ is the learnable weight matrix, and $\hat{\mathbf{D}}$ is the degree matrix associated with the adjacency matrix $\hat{\mathbf{A}}$. This operation ensures that each node in the graph aggregates feature information from its neighbors, weighted by the structural similarities encoded in $\hat{\mathbf{A}}$. This propagation mechanism allows the network to exploit contextual information across the batch, improving its ability to segment complex anatomical structures in medical images, where the relationships between different regions are critical for accurate segmentation.

### 2.3.2. Graph Neural Network Clustering

To further cluster the same graph nodes, for each 3D image volume in a mini-batch, we first extract deep features, resulting in a feature tensor $\mathcal{F} \in \mathbb{R}^{(B \times C \times W \times H \times D)}$, where $B$ is the batch size. Each voxel in the 3D volume serves as a graph node, with the feature dimensions $(W, H, D)$ representing the spatial extent of the nodes. To capture relationships across the volume, we construct a graph where each node represents a voxel and edges connect spatially adjacent nodes or nodes with high feature similarity.

Specifically, the matrix $\mathcal{W}$ for the graph is derived based on spatial and semantic affinity between patches, calculated as follows:

$$\mathcal{W} = \mathcal{F} \cdot \mathcal{F}^{\top} - \frac{Max(\mathcal{F} \cdot \mathcal{F}^{\top})}{\tau} \tag{15}$$

where $\mathcal{F}$ is the feature matrix and $\tau$ controls clustering sensitivity. $\tau$ is used to adapt the cluster selection process in correlation clustering. Since the number of clusters cannot be directly selected in correlation clustering, this parameter allows us to control the sensitivity of the process, where higher values of $\tau$ correspond to more clusters. This graph construction preserves important volumetric features, enabling the GNN to recognize spatial and semantic relationships within the medical data.

Let $\hat{\mathbf{N}}$ denote as the node feature matrix, which is derived by applying one or more layers of Graph Neural Network (GNN) convolution on a graph $G$ defined by an adjacency matrix $\mathcal{W}$. Here, we use a single-layer Graph Convolutional Network (GCN) to perform the feature extraction. The adjacency matrix $\mathcal{W}$ is constructed from the patch-wise correlation matrix based on features obtained from encoder. These correlations capture the relational structure of the patches within the visual data, thereby enabling the GNN

to leverage spatial dependencies. Formally, the GNN layer maps the input node features $N$ into a refined node feature matrix $\hat{\mathbf{N}}$ by learning the underlying data structure:

$$\hat{\mathbf{N}} = GNN(N, W; \Theta_{GNN}) \tag{16}$$

where $\Theta_{GNN}$ represents the trainable parameters of the GNN layer. Following the GNN, we utilize a Multi-Layer Perceptron (MLP) with a softmax activation applied to $\hat{\mathbf{N}}$ to produce the final output $\mathbf{S}$, which is the cluster assignment matrix. Each row of $\mathbf{S}$ corresponds to a node and represents the probability distribution over clusters, essentially encoding the likelihood of each node belonging to a particular cluster:

$$\mathbf{S} = MLP(\hat{\mathbf{N}}; \Theta_{MLP}) \tag{17}$$

where $\Theta_{MLP}$ are the MLP's trainable parameters. The optimization of the GNN model is driven by a clustering objective, with a loss function proposed to enforce distinct clustering properties.

We employ a correlation clustering loss in this work, which directly promotes intra-cluster coherence and inter-cluster separation. This loss is defined as:

$$\mathcal{L}_{\mathrm{CC}} = -\operatorname{Tr}(\mathcal{W}\mathbf{S}\mathbf{S}^T) \tag{18}$$

where $\mathcal{W}$ is redefined according to the specific correlation clustering requirements. This loss encourages nodes with high similarity (as per $\mathcal{W}$) to be assigned to the same cluster (positive affinities), while penalizing connections between dissimilar nodes (negative affinities). Consequently, this approach is advantageous for scenarios where clusters have distinct internal structures or where cluster boundaries are less clearly defined.

At each training iteration, we update the parameters $\Theta_s$ in the student network by stochastic gradient descent with the loss function(based on Eq.(10)):

$$\mathcal{L}_{\mathrm{all}} = \mathcal{L}^{\mathrm{in}} + \mathcal{L}^{\mathrm{out}} + \kappa * \mathcal{L}_{\mathrm{CC}} \tag{19}$$

We use a weight $\kappa$ to control the contribution of graph clustering to the loss function. Afterwards, teacher network parameters $\Theta_t^{(k+1)}$ at the $(k+1)$-th iteration are updated.

## 3. Experiments

### 3.1. Datasets and Evaluation Metrics

All experiments are performed on three public datasets with different imaging modalities and segmentation tasks: Automatic Cardiac Diagnosis Challenge dataset (ACDC) (Bernard et al., 2018), Atrial Segmentation Challenge dataset (LA) (Xiong et al., 2021) and Pancreas-NIH dataset (Roth et al., 2015). Four metrics are used for evaluation, including the Dice Score (%), Jaccard Score (%), 95% Hausdorf Distance (95HD), and the average surface distance

| Method | Scans used | | Metrics | | | |
|---|---|---|---|---|---|---|
| | Labeled | Unlabeled | Dice↑ | Jaccard↑ | 95HD↓ | ASD↓ |
| V-Net (Milletari et al., 2016) | 4(5%) | 0 | 52.55 | 39.69 | 47.05 | 9.87 |
| V-Net (Milletari et al., 2016) | 8(10%) | 0 | 82.74 | 71.72 | 13.35 | 3.26 |
| V-Net (Milletari et al., 2016) | 80(All) | 0 | 91.47 | 84.36 | 5.48 | 1.51 |
| UA-MT (Yu et al., 2019) | | | 82.26 | 70.98 | 13.71 | 3.82 |
| SASSNet (Li et al., 2020b) | | | 81.60 | 69.63 | 16.16 | 3.58 |
| DTC (Luo et al., 2021a) | | | 81.25 | 69.33 | 14.90 | 3.99 |
| URPC (Luo et al., 2021b) | 4(5%) | 76(95%) | 82.48 | 71.35 | 14.65 | 3.65 |
| MC-Net (Wu et al., 2021) | | | 83.59 | 72.36 | 14.07 | 2.70 |
| SS-Net (Wu et al., 2022) | | | 86.33 | 76.15 | 9.97 | 2.31 |
| BCP (Bai et al., 2023) | | | 87.07 | 77.42 | 8.83 | 2.15 |
| GraphCL | | | $88.80_{\uparrow 1.73}$ | $80.00_{\uparrow 2.58}$ | $7.16_{\downarrow 1.67}$ | $2.10_{\downarrow 0.05}$ |
| UA-MT (Yu et al., 2019) | | | 87.79 | 78.39 | 8.68 | 2.12 |
| SASSNet (Li et al., 2020b) | | | 87.54 | 78.05 | 9.84 | 2.59 |
| DTC (Luo et al., 2021a) | | | 87.51 | 78.17 | 8.23 | 2.36 |
| URPC (Luo et al., 2021b) | 8(10%) | 72(90%) | 86.92 | 77.03 | 11.13 | 2.28 |
| MC-Net (Wu et al., 2021) | | | 87.62 | 78.25 | 10.03 | 1.82 |
| SS-Net (Wu et al., 2022) | | | 88.55 | 79.62 | 7.49 | 1.90 |
| BCP (Bai et al., 2023) | | | 89.39 | 80.92 | 7.26 | 1.76 |
| GraphCL | | | $90.24_{\uparrow 0.85}$ | $82.31_{\uparrow 1.39}$ | $6.42_{\downarrow 0.84}$ | $1.71_{\downarrow 0.05}$ |

*Table 1.* Comparisons with state-of-the-art semi-supervised segmentation methods on LA dataset. Improvements compared with the second best results are emphasized in bold.

(ASD). Given two object regions, Dice and Jaccard mainly compute the percentage of overlap between them, 95HD measures the closest point distance between them and ASD computes the average distance between their boundaries. We have highlighted the results in bold when our proposed method outperforms the original counterparts.

### 3.2. Implementation Details

All experiments use default settings of $\alpha = 0.5$, $\kappa = 0.01$ and $\tau = 2$, with fixed random seeds. LA Dataset experiments run on an NVIDIA A800 GPU, while Pancreas-NIH and ACDC datasets use an NVIDIA 3090 GPU.

**LA dataset.** Following SS-Net (Wu et al., 2022), we apply rotation and flip augmentations. Training uses SGD with an initial learning rate of 0.01, decaying by 10% every 2.5K iterations. We adopt a 3D V-Net backbone, with patches cropped to $112 \times 112 \times 80$ and the size of the zero-value region of mask $\mathcal{M}$ is $74 \times 74 \times 53$. Batch size is 8, split equally between labeled and unlabeled patches, with pre-training and self-training at 5K and 15K iterations, respectively.

**ACDC dataset.** Consistent with SS-Net (Wu et al., 2022), we use a 2D U-Net backbone with patch sizes of $256 \times 256$ and the size of the zero-value region of mask $\mathcal{M}$ is $170 \times 170$. The batch size is 24, with pre-training and self-training at 10K and 30K iterations.

**Pancreas-NIH.** Based on CoraNet (Shi et al., 2021), data is augmented via rotation, rescaling, and flipping. A four-layer 3D V-Net is trained with Adam (Kingma, 2014), using an initial learning rate of 0.001. Cropped patches are $96 \times 96 \times 96$, with the size of zero-value regions of mask $\mathcal{M}$ is $64 \times 64 \times 64$. Batch size, pre-training, and self-training epochs are 8, 60, and 200, respectively.

| Method | Scans used | | Metrics | | | |
|---|---|---|---|---|---|---|
| | Labeled | Unlabeled | Dice↑ | Jaccard↑ | 95HD↓ | ASD↓ |
| U-Net (Ronneberger et al., 2015) | 3(5%) | 0 | 47.83 | 37.01 | 31.16 | 12.62 |
| U-Net (Ronneberger et al., 2015) | 7(10%) | 0 | 79.41 | 68.11 | 9.35 | 2.70 |
| U-Net (Ronneberger et al., 2015) | 70(All) | 0 | 91.44 | 84.59 | 4.30 | 0.99 |
| UA-MT (Yu et al., 2019) | | | 46.04 | 35.97 | 20.08 | 7.75 |
| SASSNet (Li et al., 2020b) | | | 57.77 | 46.14 | 20.05 | 6.06 |
| DTC (Luo et al., 2021a) | | | 56.90 | 45.67 | 23.36 | 7.39 |
| URPC (Luo et al., 2021b) | 3(5%) | 67(95%) | 55.87 | 44.64 | 13.60 | 3.74 |
| MC-Net (Wu et al., 2021) | | | 62.85 | 52.29 | 7.62 | 2.33 |
| SS-Net (Wu et al., 2022) | | | 65.83 | 55.38 | 6.67 | 2.28 |
| BCP (Bai et al., 2023) | | | 86.83 | 77.64 | 8.71 | 2.47 |
| GraphCL | | | **88.68↑$^{1.85}$** | **80.32↑$^{2.68}$** | **3.12↓$^{5.59}$** | **0.88↓$^{1.59}$** |
| UA-MT (Yu et al., 2019) | | | 81.65 | 70.64 | 6.88 | 2.02 |
| SASSNet (Li et al., 2020b) | | | 84.50 | 74.34 | 5.42 | 1.86 |
| DTC (Luo et al., 2021a) | | | 84.29 | 73.92 | 12.81 | 4.01 |
| URPC (Luo et al., 2021b) | 7(10%) | 63(90%) | 83.10 | 72.41 | 4.84 | 1.53 |
| MC-Net (Wu et al., 2021) | | | 86.44 | 77.04 | 5.50 | 1.84 |
| SS-Net (Wu et al., 2022) | | | 86.78 | 77.67 | 6.07 | 1.40 |
| BCP (Bai et al., 2023) | | | 88.84 | 80.61 | 4.42 | 1.38 |
| GraphCL | | | **89.31↑$^{0.47}$** | **81.33↑$^{0.72}$** | **2.10↓$^{2.32}$** | **0.66↓$^{0.72}$** |

*Table 2.* Comparisons with state-of-the-art semi-supervised segmentation methods on ACDC dataset. Improvements compared with the second best results are emphasized in bold.

| Method | Scans used | | Metrics | | | |
|---|---|---|---|---|---|---|
| | Labeled | Unlabeled | Dice↑ | Jaccard↑ | 95HD↓ | ASD↓ |
| V-Net (Milletari et al., 2016) | | | 69.96 | 55.55 | 14.27 | 1.64 |
| DAN (Zhang et al., 2017) | | | 76.74 | 63.29 | 11.13 | 2.97 |
| ADVNET (Vu et al., 2019) | | | 75.31 | 61.73 | 11.72 | 3.88 |
| UA-MT (Yu et al., 2019) | | | 77.26 | 63.82 | 11.90 | 3.06 |
| SASSNet (Li et al., 2020b) | 12(20%) | 50(80%) | 77.66 | 64.08 | 10.93 | 3.05 |
| DTC (Luo et al., 2021a) | | | 78.27 | 64.75 | 8.36 | 2.25 |
| CoraNet (Shi et al., 2021) | | | 79.67 | 66.69 | 7.59 | 1.89 |
| BCP (Bai et al., 2023) | | | 81.12 | 68.81 | 8.11 | 2.34 |
| GraphCL | | | **83.15↑$^{2.03}$** | **71.42↑$^{2.61}$** | **6.87↓$^{1.24}$** | **2.12↑$^{0.22}$** |

*Table 3.* Comparisons with state-of-the-art semi-supervised segmentation methods on Pancreas-NIH dataset. Improvements compared with the second best results are emphasized in bold.

### 3.3. Comparison with State-of-the-Art

We evaluate our framework on the LA and ACDC datasets, comparing it with several state-of-the-art methods, including UA-MT (Yu et al., 2019), SASSNet (Li et al., 2020b), DTC (Luo et al., 2021a), URPC (Luo et al., 2021b), MC-Net (Wu et al., 2021), and SS-Net (Wu et al., 2022). Additionally, for the LA dataset, we include comparisons with V-Net (Milletari et al., 2016), while for the ACDC dataset, we compare with U-Net (Ronneberger et al., 2015). Following the protocol in SS-Net, we conduct semi-supervised experiments with different labeled data ratios (*i.e.*, 5% and 10%). For Pancreas-NIH dataset, we evaluate with a labeled ratio of 20% (Luo et al., 2021a; Shi et al., 2021). We benchmark our method, denoted as GraphCL, against various state-of-the-art models, including V-Net (Milletari et al., 2016), DAN (Zhang et al., 2017), ADVENT (Vu et al., 2019), UA-MT (Yu et al., 2019), SASSNet (Li et al., 2020b), DTC (Luo et al., 2021a), and CoraNet (Shi et al., 2021).

**LA dataset.** To ensure a fair comparison, we adopt the identical experimental setup used in SS-Net. As shown in Table 1, our approach achieves superior performance across all four evaluation metrics, completely outperforming competing approaches. Specifically, when the labeled ratio is set to 10%, GraphCL outperforms the second-best approach by an average of 3.85% across all four evaluation metrics. With the labeled ratio of 5%, we maintain a strong advantage, showing an average improvement of 6.64 % over the second-best results across these metrics. This suggests that when the number of labeled volume is particularly limited, the knowledge from labeled data can be more effectively transferred to the unlabeled data. This phenomenon likely explains the superior performance gains observed when the labeled ratio is set to 5%. This observation also holds true for the ACDC dataset.

**ACDC dataset.** We also adopt the identical experimental setup used in SS-Net. The averaged performance results are shown in Table 2 on the ACDC dataset for four-class segmentation. Our approach consistently outperforms all state-of-the-art methods across all evaluation metrics. With the labeled ratio is set to 10%, GraphCL outperforms the second-best approach by an average of 26.01% across all four evaluation metrics. With the labeled ratio of 5%, GraphCL outperforms an average improvement of 29.65% over the second-best results across these metrics. Our approach leverages graph-based representations within the encoder and incorporates a graph neural network clustering loss $\mathcal{L}_{CC}$, which significantly contributes to the performance gains. Specifically, the integration of graph structures enables the encoder to capture complex spatial relationships and contextual dependencies among voxels, facilitating a more holistic understanding of the input data. $\mathcal{L}_{CC}$ encourages similar voxels to be grouped together while enforcing separation between distinct regions, thereby enhancing intra-cluster coherence and inter-cluster separability. Specifically, as can be seen in Figure 6, GraphCL can segment the fine details of the target organ, especially the edge details that are easily misrecognized or missed.

**Pancreas-NIH Dataset.** For the Pancreas-NIH dataset, we benchmark GraphCL against DAN (Zhang et al., 2017), ADVENT (Vu et al., 2019), UA-MT (Yu et al., 2019), SASSNet (Li et al., 2020b), DTC (Luo et al., 2021a), and CoraNet (Shi et al., 2021), all trained in a semi-supervised setup using both labeled and unlabeled data. V-Net is used as the backbone for our model and baseline methods, while V-Net alone is trained in a fully supervised manner as a lower bound. Table 3 shows that our approach achieves substantial improvements in Dice, Jaccard, and 95HD metrics, outperforming the second-best method by 2.50%, 3.79%, and 0.72%, respectively.

### 3.4. Ablation Studies

To analyze the effectiveness of each component in our proposed framework GraphCL, we conduct a series of ablation studies across three datasets (LA, ACDC, and Pancreas-NIH) with varying labeled data ratios. The detailed results

| Baseline | SA | $\mathcal{L}_{CC}$ | LA | | | | | | ACDC | | | | | |
|---|---|---|---|---|---|---|---|---|---|---|---|---|---|---|
| | | | Scans used | | Metrics | | | | Scans used | | Metrics | | | |
| | | | Labeled | Unlabeled | Dice↑ | Jaccard↑ | 95HD↓ | ASD↓ | Labeled | Unlabeled | Dice↑ | Jaccard↑ | 95HD↓ | ASD↓ |
| ✔ | ✗ | ✗ | | | 87.07 | 77.42 | 8.83 | 2.15 | | | 86.83 | 77.64 | 8.71 | 2.47 |
| ✔ | ✔ | ✗ | | | 88.13 | 78.93 | 7.46 | **1.93** | | | 88.13 | 79.46 | 5.26 | 1.45 |
| ✔ | ✗ | ✔ | 4(5%) | 76(95%) | 88.34 | 79.24 | 8.68 | 2.27 | 3(5%) | 67(95%) | 87.80 | 78.93 | **2.58** | 0.93 |
| ✔ | ✔ | ✔ | | | **88.80** | **80.00** | **7.16** | 2.10 | | | **88.68** | **80.32** | 3.12 | **0.88** |
| ✔ | ✗ | ✗ | | | 89.39 | 80.92 | 7.26 | 1.76 | | | 88.84 | 80.61 | 4.42 | 1.38 |
| ✔ | ✔ | ✗ | | | 90.00 | 81.88 | 6.87 | 1.74 | | | 89.52 | 81.61 | 3.27 | 0.97 |
| ✔ | ✗ | ✔ | 8(10%) | 72(90%) | 88.79 | 80.05 | 8.24 | 2.19 | 7(10%) | 63(90%) | 89.53 | 81.68 | 2.98 | 0.89 |
| ✔ | ✔ | ✔ | | | **90.24** | **82.31** | **6.42** | **1.71** | | | 89.31 | 81.33 | **2.10** | **0.66** |

Table 4. Ablation study. The best results are emphasized in bold.

| Baseline | SA | $\mathcal{L}_{CC}$ | Pancreas-NIH | | | | | |
|---|---|---|---|---|---|---|---|---|
| | | | Scans used | | Metrics | | | |
| | | | Labeled | Unlabeled | Dice↑ | Jaccard↑ | 95HD↓ | ASD↓ |
| ✔ | ✗ | ✗ | | | 81.12 | 68.81 | 8.11 | 2.34 |
| ✔ | ✔ | ✗ | | | 82.47 | 70.53 | 7.15 | 2.42 |
| ✔ | ✗ | ✔ | 12(20%) | 50(80%) | 82.32 | 70.34 | 10.56 | 3.63 |
| ✔ | ✔ | ✔ | | | **83.15** | **71.42** | **6.87** | **2.12** |

Table 5. Ablation study. The best results are emphasized in bold.

| Layers | Scans used | | Metrics | | | |
|---|---|---|---|---|---|---|
| | Labeled | Unlabeled | Dice↑ | Jaccard↑ | 95HD↓ | ASD↓ |
| 1 | | | 0.030 | 0.016 | 80.51 | 45.53 |
| 2 | | | 72.59 | 61.30 | 17.12 | 5.22 |
| 3 | 3(5%) | 67(95%) | 87.83 | 79.04 | 3.24 | 0.98 |
| 4 | | | 88.05 | 79.35 | 4.17 | 1.19 |
| 5 | | | **88.68** | **80.32** | **3.12** | **0.88** |
| 1 | | | 0.008 | 0.004 | 40.94 | 23.73 |
| 2 | | | 72.35 | 59.20 | 15.14 | 4.32 |
| 3 | 7(10%) | 63(90%) | 88.72 | 80.40 | 4.01 | 1.15 |
| 4 | | | 89.10 | 80.99 | 5.94 | 1.43 |
| 5 | | | **89.31** | **81.33** | **2.10** | **0.66** |

Table 6. Ablation study on ACDC. The best results are emphasized in bold.

are presented in Table 4, Table 5, and Table 6.

**Effectiveness of Components.** In these experiments, we examine the contribution of two key components: the structure-aware alignment (denoted as SA) and the graph neural network clustering loss ($\mathcal{L}_{CC}$). As shown in Table 4, adding SA or $\mathcal{L}_{CC}$ individually improves performance compared to the baseline. Specifically, incorporating both SA and $\mathcal{L}_{CC}$ achieves the best results, with notable improvements in metrics such as Dice Score, Jaccard Index, 95HD, and ASD. For instance, on the LA dataset with a 10% labeled data ratio, GraphCL with both components achieves a Dice Score of 90.24%, outperforming the baseline by a substantial margin.

**Optimal Placement of GCN Layers.** We further investigate the impact of placing the Graph Convolutional Network (GCN) layers at different depths within the encoder, as shown in Table 6. The results indicate that inserting the

GCN layers at deeper levels (specifically at the fifth layer) leads to the highest performance gains. For example, with a labeled ratio of 10% on the ACDC dataset, inserting the GCN at the fifth layer results in a Dice Score of 89.31% and an ASD of 0.66, which are significantly better than inserting GCN at shallower layers. This demonstrates that deeper placement of GCNs allows the model to capture more complex spatial dependencies and contextual information, thereby enhancing segmentation accuracy.

**Dataset-Specific Observations.** Each dataset demonstrates unique performance patterns based on the labeled data ratio and the presence of SA and $\mathcal{L}_{CC}$. For example, on the Pancreas-NIH dataset (Table 5), combining SA and $\mathcal{L}_{CC}$ results in improvements of 2.50% in Dice Score and 0.72 in 95HD over the baseline. The clustering loss $\mathcal{L}_{CC}$ contributes substantially to boundary delineation by enforcing inter-cluster separability, which is particularly beneficial for segmenting complex anatomical structures.

These ablation studies highlight the importance of each component in GraphCL, demonstrating that the combination of structure-aware alignment and graph-based clustering significantly improves segmentation results across various medical image datasets.

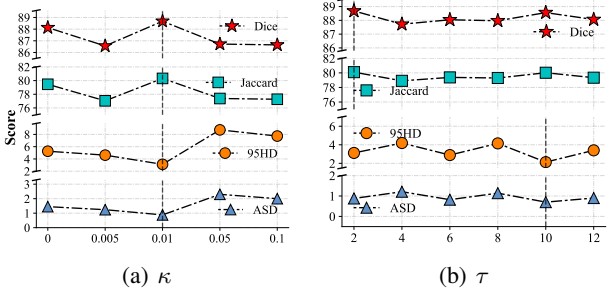

Figure 3. Sensitivity analysis on the ACDC dataset with the labeled ratio of 3(5%).

### 3.5. Impacts of Hyper-parameters

To further verify the effectiveness of the proposed method, we also conduct sensitivity analysis on the ACDC dataset

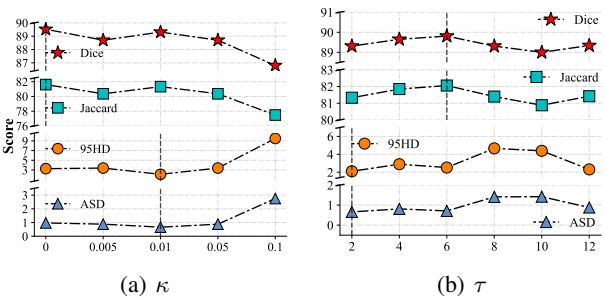

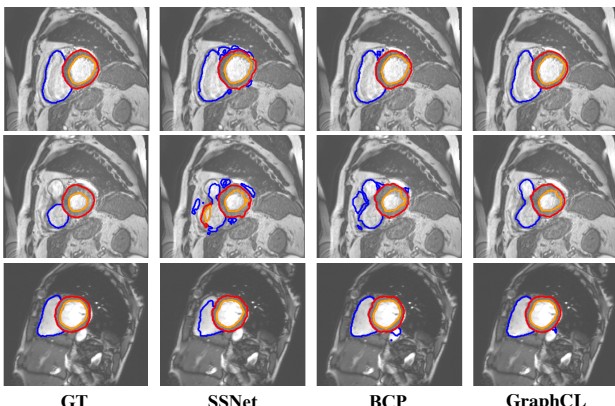

(a) $\kappa$          (b) $\tau$

*Figure 4.* Sensitivity analysis on the ACDC dataset with the labeled ratio of 7(10%) .

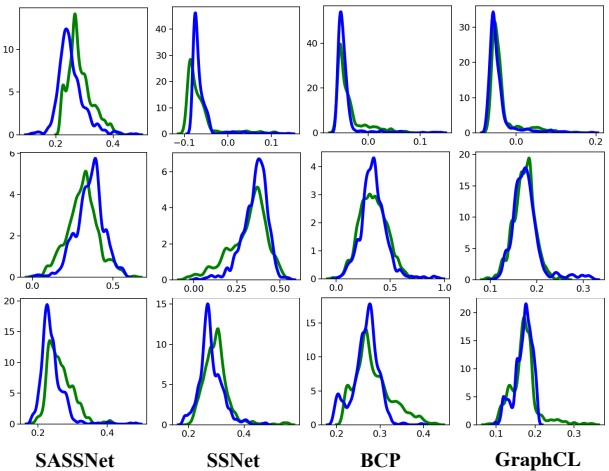

SASSNet      SSNet      BCP      GraphCL

*Figure 5.* Kernel dense estimations of different methods, trained on 5% labeled ACDC dataset. Top to bottom are kernel-dense estimations of features belonging to three different classes of ACDC: left ventricle, myocardium, and right ventricle.

to evaluate the impact of hyperparameters $\kappa$ and $\tau$, with labeled data ratios of 5% and 10% (Figure 3 and Figure 4).

**Effect of $\kappa$ (Figure 3 (a) and Figure 4 (a)).** $\kappa$ controls the weight of the structure-aware alignment. Increasing $\kappa$ initially improves the Dice and Jaccard scores, reaching a peak around 0.01, after which performance declines slightly. This indicates that a moderate $\kappa$ achieves the best balance between alignment and segmentation quality.

**Effect of $\tau$ (Figure 3 (b) and Figure 4 (b)).** $\tau$ controls clustering sensitivity for the clustering loss $\mathcal{L}_{CC}$. $\tau = 2$ yields the most consistent overall performance across multiple metrics, with Dice, Jaccard, 95HD, and ASD showing stable, favorable results. Although there are instances where certain metrics reach peak values at $\tau = 6$ and $\tau = 10$, the variation in results between these values and $\tau = 2$ is relatively small. Therefore, $\tau = 2$ can be considered an optimal choice, as it offers reliable performance without significant trade-offs, making it a robust setting for clustering sensitivity.

GT      SSNet      BCP      GraphCL

*Figure 6.* Visualizations of several semi-supervised segmentation methods with 5% labeled data and ground truth on ACDC dataset. The blue, red, and orange lines represent the 25%, 50%, and 75% locations of the segmented area, respectively.

## 3.6. Visualization Analysis

Figure 5 and Figure 6 display presents kernel density estimations and segmentation results for different methods trained on the ACDC dataset, trained with 5% labeled data. In Figure 5, among all three cardiac structures, GraphCL has the best alignment of feature distributions between labeled and unlabeled data. This is evident when compared to SASSNet, SSNet, and BCP. Figure 6 illustrates that the segmentation results from GraphCL are notably more accurate and precise. In contrast to SSNet and BCP, GraphCL presents tighter and more distinct boundaries, demonstrating a closer alignment with the ground truth (GT). These findings underscore GraphCL's superior capability in capturing critical features and enhancing segmentation performance, particularly in scenarios with limited labeled data.

## 4. Conclusion

In this paper, we propose a novel graph-based clustering for semi-supervised medical image segmentation (SSMIS) that models graph data structures within a unified framework. Our approach leverages CNN-derived features from samples to construct a densely connected instance graph, based on the structural similarity, which effectively captures semantic representations for SSMIS. Additionally, we introduce a graph clustering mechanism to utilize more information during the clustering process, enabling implicit semantic part segmentation. Extensive experiments demonstrate the effectiveness of our proposed approach. For future work, we will explore methods to generate more reliable labels and enhance graph accuracy, aiming to reduce noise within the input graph.

## Impact Statement

This paper presents work whose goal is to advance the field of Semi-supervised Image Segmentation. There are many potential societal consequences of our work, none which we feel must be specifically highlighted here.

## Acknowledge

This work is supported by the National Natural Science Foundation of China under Grants No. 62406100, Tianjin Natural Science Foundation under Grants No. 24JC-QNJC00320, Beijing Postdoctoral Research Foundation. This work is also supported by funding from the Hong Kong RGC General Research Fund (No. 152211/23E, 15216424/24E, and 152115/25E), the PolyU Internal Fund (No. P0056171), and the Huawei Gifted Fund.

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
