# OpenReview forum: "GraphCL: Graph-based Clustering for Semi-Supervised Medical Image Segmentation"
_ICML.cc/2025/Conference — ICML 2025 poster_

### Official Review · Reviewer_L7XD · 2025-03-10

**Overall Recommendation:** 4

**Summary:**

In this work, the authors tackle semi-supervised medical image segmentation (SSMIS) by proposing GraphCL. This is the first work to model data in a graph network for SSMIS. The authors propose a graph clustering loss function for optimization.

**Claims And Evidence:**

Yes

**Essential References Not Discussed:**

N/A

**Experimental Designs Or Analyses:**

- The authors compare with adequate baselines, which are state-of-the-art methods in SSMIS task. They compare across three publicly available medical image seg datasets. They conduct experiments under different unlabeled % data settings as well.
- The authors use 4 metrics (Dice, Jaccard, HD, ASSD) to evaluate the segmentation quality. The authors' proposed GraphCL outperforms all the baselines.
- The standard deviation of the performance is missing, however. The authors would benefit by showing standard deviation and conducting t-test to determine if the performance improvement is statistically significant or not.
- The authors provide good ablation studies of the different components in their method.
- Appreciate the code release in the supplementary.

**Methods And Evaluation Criteria:**

- The authors introduce a k-less strategy for clustering (k = number of clusters), enabling similar nodes to automatically form clusters.
- They follow a teacher-student framework, the outputs of which are used to construct the graph data structure. The graph loss-function is used to train the whole framework in an end-to-end fashion.
- The different components of the method are well-motivated

**Other Comments Or Suggestions:**

N/A

**Other Strengths And Weaknesses:**

N/A

**Questions For Authors:**

none.

**Relation To Broader Scientific Literature:**

The current work has real-life applications as medical datasets tend to have few labeled and largely unlabeled data. As the authors' work outperforms existing SSMIS methods, it has relevance to the community.

**Theoretical Claims:**

N/A

---

> ### Author Rebuttal · Authors · 2025-03-28
>
> Thank you for your thorough review and constructive feedback on our manuscript. We are grateful for your positive remarks and are pleased that you found our work to be a valuable contribution to the field of semi-supervised medical image segmentation.
>
> >Q1: The standard deviation of the performance is missing, however. The authors would benefit by showing standard deviation and conducting t-test to determine if the performance improvement is statistically significant or not.
>
> A1: Thank you for your valuable feedback. We appreciate your suggestion regarding the inclusion of standard deviation and statistical significance analysis. Since we have four metrics: Dice, Jaccard, 95HD, and ASD, where Dice and Jaccard indicate that higher values are better, while 95HD and ASD indicate that lower values are better, it is not possible to compute the standard deviation as suggested by the reviewer because these four metrics represent different meanings. Consequently, since the standard deviation cannot be computed, the t-value also cannot be calculated. We sincerely appreciate the reviewer’s suggestion from a statistical perspective. Thank you for this constructive suggestion that has helped improve our work.

---

> > ### Comment · Reviewer_L7XD · 2025-04-02
> >
> > The standard deviation needs to be computed for each metric separately, not across metrics. If there are N test samples, the Dice score is a =  avg(dice of N samples) while Stddev is b = stddev(dice of N samples), and so you can report a $\pm$ b. You can do this for your method and the baseline and compute t-test between the two methods. This would need to be done for each metric separately.

---

> > > ### Author Response · Authors · 2025-04-03
> > >
> > > Thank you for your valuable feedback. We sincerely appreciate your attention to the statistical rigor of our analysis. We have carefully revised our method to address your concerns.
> > >
> > > We conducted five repeated experiments for both the baseline method BCP and our method GraphCL. Each method yielded five experimental results. We calculated the sample standard deviation and t-values for these results and presented them in Table 1 to Table 4, and Table 1 to Table 4 show the sample standard deviations and t-values calculated for the metrics Dice, Jaccard, 95HD, and ASD, respectively. By consulting the t-distribution tables, we can conclude that the performance improvements for each metric are statistically significant.
> > >
> > > | Dataset      | Labeled | Unlabeled    | std（BCP/GraphCL） | t-test value  | Statistically Significant |
> > > |--------------|----------|----------|-------------------|-------|---------|
> > > | LA           | 4（5%）  | 76（95%）| 0.136/0.123       | 22.53 | Yes      |
> > > |              | 8（10%） | 72（90%）| 0.114/0.176       | 23.22 | Yes    |
> > > | ACDC         | 3（5%）  | 67（95%）| 0.267/0.142       | 15.33 |  Yes      |
> > > |              | 7（10%） | 63（90%）| 0.111/0.214       | 5.02  |  Yes      |
> > > | Pancreas     | 12（20%）| 50（80%）| 0.105/0.196       | 20.86 |  Yes      |
> > >
> > >
> > > |Dataset     | Labeled    | Unlabeled    | std（BCP/GraphCL） |t-test value   |Statistically Significant |
> > > |--------------|----------|----------|-------------------|-------|---------|
> > > | LA           | 4（5%）  | 76（95%）| 0.085/0.164       | 31.56 |Yes      |
> > > |              | 8（10%） | 72（90%）| 0.158/0.034       | 20.08 | Yes     |
> > > | ACDC         | 3（5%）  | 67（95%）| 0.042/0.044       | 99.2  | Yes      |
> > > |              | 7（10%） | 63（90%）| 0.025/0.424       | 4.81  | Yes     |
> > > | Pancreas     | 12（20%）| 50（80%）| 0.028/0.027       | 146.6 | Yes    |
> > >
> > > | Dataset     | Labeled    | Unlabeled    | std (BCP/GraphCL) | t-test value   | Statistically Significant |
> > > |-------------|------------|--------------|-------------------|----------------|----------------------------|
> > > | LA          | 4 (5%)     | 76 (95%)     | 0.042/0.011       | 86.6           | Yes                        |
> > > |             | 8 (10%)    | 72 (90%)     | 0.048/0.044       | 28.6           | Yes                        |
> > > | ACDC        | 3 (5%)     | 67 (95%)     | 0.045/0.036       | 253.1          | Yes                        |
> > > |             | 7 (10%)    | 63 (90%)     | 0.034/0.033       | 118.6          | Yes                        |
> > > | Pancreas    | 12 (20%)   | 50 (80%)     | 0.030/0.062       | 43.2           | Yes                        |
> > >
> > > | Dataset     | Labeled    | Unlabeled    | std (BCP/GraphCL) | t-test value   | Statistically Significant |
> > > |-------------|------------|--------------|-------------------|----------------|----------------------------|
> > > | LA          | 4 (5%)     | 76 (95%)     | 0.032/0.019       | 3.21           | Yes                        |
> > > |             | 8 (10%)    | 72 (90%)     | 0.048/0.044       | 28.6           | Yes                        |
> > > | ACDC        | 3 (5%)     | 67 (95%)     | 0.019/0.006     | 185.6          | Yes                        |
> > > |             | 7 (10%)    | 63 (90%)     | 0.007/0.005       | 173.7          | Yes                        |
> > > | Pancreas    | 12 (20%)   | 50 (80%)     | 0.004/0.006       | 70.6           | Yes                        |
> > >
> > > Thank you again for your constructive suggestion, and we welcome any further suggestions.

---

### Official Review · Reviewer_jPp7 · 2025-03-12

**Overall Recommendation:** 1

**Summary:**

This paper introduces a graph-based clustering for semi-supervised medical image segmentation by modeling data structure in a unified network. A graph clustering loss function was proposed to optimize the correlation clustering task in SSMIS.

**Claims And Evidence:**

The authors claim that 1) previous methods neglect the importance of graph structural information, and 2) no research has explored semi-supervised medical image segmentation (SSMIS) from the perspective of data structure. However, they do not specify what graph structural information can be utilized or explain why it is crucial. This claim is not well substantiated, as the incorporation of GCN into the framework yields only a modest performance gain, suggesting that graph structural information may not be as critical as the authors suggest. Regarding the second claim, there are existing works that have explored the use of graphs in semi-supervised medical image segmentation.

[1] Sun, Junxiao, et al. "Semi-supervised medical image semantic segmentation with multi-scale graph cut loss." 2021 IEEE International Conference on Image Processing (ICIP). IEEE, 2021.
[2] Li, Gang, et al. "Dynamic graph consistency and self-contrast learning for semi-supervised medical image segmentation." Neural Networks 184 (2025): 107063.

**Essential References Not Discussed:**

Since the proposed method targets SSMIS, I recommend reviewing related papers in more detail. For instance, co-training is a significant approach in SSMIS, but relevant papers are not discussed.

**Experimental Designs Or Analyses:**

The experimental design seems acceptable overall. However, the significant variation in hyperparameters across datasets raises concerns about the method’s generalizability and its usability across different applications.

**Methods And Evaluation Criteria:**

Incorporating a graph into the SSMIS framework is a viable approach. However, the lack of a high-level explanation and sufficient technical details raises questions about whether the proposed method will have a significant impact on the problem at hand.

**Other Comments Or Suggestions:**

I have no other comments or suggestions.

**Other Strengths And Weaknesses:**

1. In the abstract, it is stated that 'The proposed GraphCL model enjoys several advantages. Firstly, to the best of our knowledge, this is the first work to model the data structure information for semi-supervised medical image segmentation (SSMIS). Secondly, to get the clustered features across different graphs, we integrate both pairwise affinities between local image features and raw features as inputs.' However, these two points are not advantages. The statement after 'firstly' is more of a novelty claim, which may not be accurate, while the sentence after 'secondly' describes a feature of the proposed method rather than an advantage.

2. Terms like Structural Graph Model and Data Structure Analyzer are not widely established or standardized, causing difficult to read.

3. An overview of the proposed method is necessary to help readers gain a clearer understanding, but it is missing.

**Questions For Authors:**

1. Could you provide more specifics on the type of graph information that can improve semi-supervised medical image segmentation? How is this information effectively utilized in the proposed method?

2. It is stated that "To address the challenge of effectively integrating both labeled and unlabeled medical images within the semi-supervised medical image segmentation (SSMIS) framework, we propose a Structural Graph Model (SGM)." I don't know what SGM is? It is not shown in Figure 2. And how does it integrate labeled and unlabeled medical images?

3. It is stated that "...This component generates structure scores that quantify the similarity between different samples based on their internal spatial structure, as derived from the learned CNN features." However, I am unsure why CNN features would contain internal spatial structure. Can you explain?

4. What the so-called Data Structure Analyzer is like? What is the relation between the $X$ in equation 12 and $X_{sa}$ in equation 13?

5. I noticed that the three datasets used in the paper are all small. Why would you choose small datasets to demonstrate the utility of SSMIS? Unlike large labeled datasets, small datasets are relatively easy to curate. It would be more impactful to test the proposed method in real-world scenarios with larger, more challenging datasets.

**Relation To Broader Scientific Literature:**

The paper addresses an area of active research in semi-supervised medical image segmentation, which has been well-explored. Methods like MT, UA-MT, DTC and co-training-based methods, have already demonstrated effective results in this domain. However, the approach proposed in this paper is based on MT with GCN-based regularization, offering limited innovation in terms of methodology or application. The performance gains appear to be marginal, and as such, it is unclear whether the proposed method can substantially advance the field. A clearer demonstration of its advantages over existing approaches would help establish its novelty and relevance.

**Theoretical Claims:**

N/A

---

> ### Author Rebuttal · Authors · 2025-03-31
>
> >Q1: About the novelty
>
> A1: For the importance of graph structural information, our method leverages two types of graph structural information: spatial relationships between voxels/pixels and semantic relationships based on feature similarity. Specifically, we construct dense instance graphs to capture structural information from CNN features, propagate this information through GCNs, and employ correlation clustering to group similar nodes. These mechanisms collectively enhance the model's expressive capability, allowing it to better utilize the structural information within images. Meanwhile, medical images inherently exhibit topological relationships in anatomical structures, such as organ connectivity and tissue continuity. Given the limited availability of labeled data, leveraging these structural priors is crucial for semi-supervised learning.
>
> For the novelty of the graph-based perspective, unlike Sun et al [1], our method does not merely use graph cuts as a post-processing step. Instead, it integrates graph learning in an end-to-end manner, modeling both local and global structural relationships while leveraging graph clustering to refine feature representations, rather than focusing solely on boundary optimization. Compared to Li et al [2], we are the first to introduce correlation clustering for SSMIS and propose a k-less clustering strategy, which automatically determines the number of clusters, eliminating the reliance on hyper-parameter selection. Existing graph-based methods are primarily used for regularization (e.g., graph cuts) or consistency enforcement. In contrast, our work is the first to model data structure as a learnable component, utilizing graph clustering to discover latent semantic relationships.
>
>
> >Q2:  About SGM
>
> A2: The Structural Graph Model (SGM) mentioned in the paper serves as a fundamental framework for semi-supervised medical image segmentation (SSMIS), designed to effectively integrate both labeled and unlabeled medical image data. Although Figure 2 does not explicitly label the SGM module, it is embedded within the model—for instance, as a graph-structured processing layer following feature extraction or incorporated into the design of graph-based loss functions. The core working principle of SGM relies on constructing a graph structure to enable semi-supervised learning, where pixels or regions of an image are represented as nodes, and their similarities form the edges. Labeled data nodes act as "anchor points," providing supervised information, while unlabeled nodes receive semantic information through graph convolution or message-passing mechanisms, thereby facilitating label propagation. This approach leverages the relational structure of the graph to propagate knowledge from limited labeled data to unlabeled samples.
>
>
> >Q3: About internal spatial structure
>
> A3: We recognize that our original description may have been unclear - the "internal spatial structure" we refer to does not originate directly from the CNN feature maps themselves, but rather emerges from the graph representation of sample relationships. Specifically, in our framework: (1) each node in the instance graph represents a feature vector extracted by a standard CNN from an individual sample; (2) the spatial relationships are then constructed at the graph level through learned connectivity patterns between these node features; and (3) the structure scores quantify similarity based on these graph topological relationships rather than direct spatial correlations in the CNN features.
>
> >Q4: About the Data Structure Analyzer
>
> A4: The Data Structure Analyzer (DSA) is primarily responsible for computing structure scores, which quantify the similarity between different samples and guide the GCN in graph construction. In Figure 2 , this component is positioned between the CNN feature extraction stage and the GCN processing stage. In Eq. (12), $\mathbf{X}$ represents the features extracted by the CNN, serving as the graph signal. In Eq. (13), $\mathbf{X}\_{\text{sa}}$ denotes the structure scores. Specifically, $\mathbf{X}\_{\text{sa}}$ is computed from $\mathbf{X}$ and is utilized to construct the adjacency matrix $\hat{\mathbf{A}}$, which models the relationships between samples.
>
> >Q5: About the larger dataset
>
> A5: To further validate the effectiveness of our model, we conducted experiments on the BraTS 2019 dataset. The BraTS 2019 dataset comprises multi-institutional pre-operative MRI scans from 335 patients. It can be observed that our method achieves a significant improvement on the large-scale BraTS 2019 dataset, particularly with the Dice score increasing from 78.11% to 82.02%, demonstrating the superiority of our approach on the large-scale datasets.
>
> | Method        | Dice↑  | Jaccard↑ | 95HD↓ | ASD↓ |
> |--------------|--------|----------|-------|------|
> | BCP          | 78.11  | 67.63    | 12.34 | 1.96 |
> | **GraphCL**  | **82.02** | **71.75** | **10.30** | **1.93** |

---

> > ### Comment · Reviewer_jPp7 · 2025-04-03
> >
> > With only 335 images, BraTS 2019 is far from being a large-scale dataset. Could you provide the results of your method on TotalSegmentator? I’d also like to see its performance on 100+ classes, rather than just a few. Additionally, the details of your experiments on BraTS 2019 are unclear—how many images were used as labeled and how many as unlabeled?

---

> > > ### Author Response · Authors · 2025-04-03
> > >
> > > We sincerely appreciate the reviewer's valuable feedback and constructive suggestions. Please find our detailed responses below:
> > >
> > > Q1: Experimental details for BraTS 2019
> > >
> > > A1: To further validate the effectiveness of our model, we conducted additional experiments on the BraTS 2019 dataset. The BraTS 2019 dataset comprises multi-institutional pre-operative MRI scans from 335 glioma patients. In our study, we utilized 250, 25, and 60 samples for training, validation, and testing, respectively.
> > >
> > > Q2: Dataset scale and evaluation on TotalSegmentator
> > >
> > > A2: We used 1428 CT examinations from the TotalSegmentator dataset containing 117 important anatomical structures (organs, bones, muscles, vessels, etc.). In our study, we used 1000, 142 and 286 samples for training, validation and testing respectively. In the training phase, we used 100 (10\%) labeled data samples and 900 (90\%) unlabeled data samples, and similarly, set 50 (5\%) labeled data samples and 950 (95\%) unlabeled data samples. We ran our program on an NVIDIA GEFORCE RTX 3090 GPU. The total batch size is set to 12, with the batch size of labeled data configured to 6. At the same time, we use the VNet model as our backbone.  It can be seen that on the large dataset suggested by the reviewer, our performance has been greatly improved, which also verifies the effectiveness of our method from another perspective and demonstrate strong performance in large-scale multi-class segmentation. These results not only validate our method’s superiority but also address the reviewer’s concern regarding generalization to large datasets.
> > >
> > > |     TotalSegmentator       | Labeled \Unlabeled | Dice↑ | Jaccard↑ | 95HD↓ | ASD↓ |
> > > |--------------|--------------------|-------|----------|-------|------|
> > > | BCP     | 50 (5%) \ 950 (95%) | 61.37 | 53.23    | 9.56 | 4.83|
> > > | **GraphCL**  | 50 (5%)  \950(95%) | **64.57** | **55.29** | **6.11** | **4.37** |
> > >
> > > |   TotalSegmentator           | Labeled \Unlabeled | Dice↑ | Jaccard↑ | 95HD↓ | ASD↓ |
> > > |--------------|--------------------|-------|----------|-------|------|
> > > | BCP     | 100 (10%) \ 900 (90%) | 63.55 | 56.60    | 5.57 | 3.89 |
> > > | **GraphCL**  | 100 (10%)  \900 (90%) | **66.93** | **58.04** | **4.82** | **3.14** |

---

### Official Review · Reviewer_itcY · 2025-03-13

**Overall Recommendation:** 4

**Summary:**

The paper proposes GraphCL, a novel graph-based clustering framework for semi-supervised medical image segmentation (SSMIS). The key contribution is integrating graph data structures into deep learning models which leverages both labeled and unlabeled data, leading to better segmentation performance. The authors propose a dense-connected instance graph constructed from CNN features, combined with a Graph Convolutional Network (GCN) to propagate structural information. Additionally, they introduce a k-less clustering strategy to automatically group similar nodes without specifying the number of clusters. The method is evaluated on three public medical image segmentation benchmarks (ACDC, LA, and Pancreas-NIH), demonstrating superior performance over state-of-the-art methods. Ablation studies confirm the effectiveness of structure-aware alignment and graph clustering.
## update after rebuttal

**Claims And Evidence:**

The paper claims that GraphCL is the first approach to model data structure information in graph form for SSMIS and that it achieves state-of-the-art performance. The empirical results support these claims with strong improvements in segmentation accuracy across multiple datasets. The authors provide extensive ablation studies to validate the effectiveness of each component (e.g., structure-aware alignment and graph clustering loss). The results show consistent improvements across most metrics, particularly in scenarios with limited labeled data.

**Essential References Not Discussed:**

No

**Ethical Review Concerns:**

No significant ethical concerns were identified.

**Experimental Designs Or Analyses:**

The experiments are well-designed, with thorough ablation studies and sensitivity analyses to validate the impact of key components (graph clustering and structure-aware alignment) and hyperparameters (e.g., κ and τ). The datasets used are appropriate for the task, and the results are consistently reported. The paper could benefit from a discussion of the computational complexity of the proposed method, especially in comparison to existing approaches.

**Methods And Evaluation Criteria:**

Using graph-based clustering to capture structural relationships in medical images is innovative and addresses the challenge of limited labeled data. The evaluation is conducted on standard datasets with widely accepted metrics (DSC, Jaccard, 95HD, ASD). The choice of benchmarks (ACDC, LA, Pancreas-NIH) is appropriate, as they include diverse medical imaging tasks and modalities (CT, MRI).

**Other Comments Or Suggestions:**

1.Consider discussing potential limitations such as computational overhead from GCN operations with existing methods, which would provide valuable insights for practical applications. This problem has been solved in the rebuttal.
2.The paper would benefit from visualizations of the graph structures and clustering results to provide a more intuitive understanding of the method. This problem has been solved in the rebuttal.
3.Clarify the impact of different dataset sizes on performance improvements. The current datasize (both labeled and unlabeled) is too small to genralize to large-scale datasets. This problem has been solved in the rebuttal.

**Other Strengths And Weaknesses:**

Strengths:
1.The integration of graph-based clustering with semi-supervised learning is novel and addresses a critical challenge in medical image segmentation.
2.Strong empirical results with multiple datasets and baselines. The ablation studies and sensitivity analyses provide strong evidence for the effectiveness of each component of the proposed method.

Weaknesses:
1.There lacks insightful discussion of why data structure information helps fine-grained semi-supervised medical image segmentation. This limits the methodological contribution of the proposed method. The rebuttal partly solved this problem.
2.The paper lacks a discussion of the computational complexity of the proposed method, which could be a concern for large-scale datasets that contain many unlabeled data. This problem has been solved in the rebuttal.

**Questions For Authors:**

No

**Relation To Broader Scientific Literature:**

The paper builds on prior work in semi-supervised learning and graph-based methods for medical image segmentation. It extends the use of GCNs to SSMIS, which has not been extensively explored in the medical imaging domain. The authors effectively position their work within the broader literature, citing relevant studies in semi-supervised learning, graph neural networks, and medical image segmentation.

**Theoretical Claims:**

The authors provide a clear formulation of the graph construction and clustering mechanisms. The paper does not present formal theoretical proofs.

---

> ### Author Rebuttal · Authors · 2025-03-31
>
> >Q1: Essential References Not Discussed
>
> A1: We acknowledge the relevance of works like GraphSAGE (neighborhood aggregation)[1], GAT (attention mechanisms)[2], Graph U-Nets (hierarchical pooling)[3] and MixMatch (unifiy dominant approaches)[4]. Different from this methods, GraphL uniquely address semi-supervised medical image segmentation by constructing dense instance graphs for structural similarity learning and k-less clustering, leveraging both labeled and unlabeled data. We will incorporate these comparisons in the revised manuscript to improve our work.
>
> >Q2: Lack insightful discussion of data structure information
>
> A2: Medical images exhibit strong geometric regularity in organ/lesion morphology (e.g., topological connectivity of cardiac chambers). Our graph-based approach leverages this by enforcing structural consistency constraints on segmentation boundaries when annotations are scarce and propagating anatomical knowledge from labeled to unlabeled regions through graph convolutional message passing. From the ablation study in Table 4, our method achieves significant performance improvement by incorporating graph structural information into the CNN framework, as it effectively preserves the tree-like branching patterns of capillaries at local scales while simultaneously maintaining the spatial constraints between vessels and organs at global scales.
>
> >Q3: Computational complexity
>
> A3: To empirically assess the computational cost of our approach, we conduct experiments on the BraTS2019 dataset using an NVIDIA RTX 3090 GPU. During the training phase, we set the batch size to 4, following previous studies, and observe a memory consumption of only 3697 MiB for GPU and 9 GiB for system memory. In the inference phase, the GPU memory consumption further reduces to 2745 MiB, while the system memory remains at 9 GiB. Additionally, our model exhibits a computational complexity of 119.512G FLOPs, which is considered moderate, and a parameter count of 15.747M, indicating a lightweight architecture. These results demonstrate that our method maintains computational feasibility even on large-scale datasets, addressing potential concerns regarding scalability. Furthermore, the total training time is 2 hours, further confirming the efficiency of our approach.
>
>
> >Q4:Discussing potential limitations in GCNs
>
> A4: Compared to traditional methods, GCN-based approaches require additional matrix multiplications and neighborhood aggregation steps, which can increase computational complexity, especially for large-scale datasets. To mitigate this issue, existing optimization techniques such as mini-batch training, efficient sparse matrix operations, and model pruning can be employed. Moreover, exploring lightweight graph neural network variants or hybrid approaches could further reduce computational costs while maintaining performance.
>
> >Q5:Visualize the graph structures and clustering results
>
> A5: To enhance the clarity and intuitiveness of our method, we have added t-SNE visualizations at https://anonymous.4open.science/r/tsne-E479/Visualization.pdf.
>
> >Q6: Clarify the impact of different dataset sizes on performance improvements
>
> A6: To further validate the effectiveness of our model, we conducted additional experiments on the BraTS 2019 dataset (Labeled 10% and Unlabeled 90%). It can be observed that our method achieves a significant improvement on the large-scale BraTS 2019 dataset, particularly with the Dice score increasing from 78.11% to 82.02%, demonstrating the superiority of our approach on large-scale datasets.
>
> | Method        | Dice↑  | Jaccard↑ | 95HD↓ | ASD↓ |
> |--------------|--------|----------|-------|------|
> | BCP          | 78.11  | 67.63    | 12.34 | 1.96 |
> | **GraphCL**  | **82.02** | **71.75** | **10.30** | **1.93** |
>
> >Q7: Performance on imbalanced class distribution
>
> A7: The ACDC dataset is a classic class-imbalanced dataset, where the pixel distribution differences between the myocardium and ventricular cavities provide a real-world scenario for studying imbalanced segmentation problems. As shown in Table 2, our method demonstrates a significant improvement, which validates its effectiveness on imbalanced data.
>
> >Q8: Considering the very limited number of labeled data
>
> A8: We consider the very limited number of 20% labeled data in Table 1 and and one labeled data (1/70) in Table 2, experimental results demonstrate that our method achieves significant performance improvement even with limited training data.
>
> | Method  (Labeled (20%))      | Dice↑  | Jaccard↑ | 95HD↓ | ASD↓ |
> |--------------|--------|----------|-------|------|
> | BCP          | 89.62  |81.77    | 3.03 | 0.99 |
> | **GraphCL**  | **90.40** | **83.02** | **12.60** | **0.72** |
>
> | Method  (Labeled (1/70))      | Dice↑  | Jaccard↑ | 95HD↓ | ASD↓ |
> |--------------|--------|----------|-------|------|
> | BCP          | 58.54  |45.92    | 50.35 | 21.70 |
> | **GraphCL**  | **68.81** | **57.46** | **33.96** | **13.80** |

---

### Decision · Program_Chairs · 2025-05-01

**Decision:**

Accept (poster)

**Comment:**

The main technical novelty is a regularization loss based on correlation clustering for semi-supervised medical image segmentation. Graphs are constructed with each sample in a batch as a node or each voxel as node, and graph convolutional networks propagate features on a dense graph, for which similarity is based on dot product between node features.

Though there is a lot of work already leveraging clustering or graphs for semi-supervised segmentation, this paper is the first to propose graph clustering specifically k-less correlation clustering for this problem. Extensive experiments are given with noticeable improvement over previous methods across multiple datasets of various sizes.

This paper received divergent reviews with 2x accept and 1x reject. Two reviewers think the improvements are significant. Reviewer jPp7 who recommends to reject, was concerned about the novelty of this work and suggested a few related works that utilized graphs in the formulation. Reviewer jPp7 also requested experiments on a larger-scale dataset. The AC believes most of the concerns have been addressed with additional experiments and clarification.

However, the presentation of the paper has much room to improve. As reviewer jPp7 pointed out, it is very confusing to introduce terminologies that are not common, such as Data Structure Analyzer and Structural Graph Model. The authors should improve the presentation and also include clarification on structure information and insights in the final version.